# Resolving the mechanisms of hygroscopic growth and cloud condensation nuclei activity for organic particulate matter

Pengfei Liu[1], Mijung Song[2,3], Tianning Zhao[1], Sachin S. Gunthe[1,4], Suhan Ham[3], Yipeng He[1,5], Yi Ming Qin[1], Zhaoheng Gong[1], Juliana C. Amorim[1], Allan K. Bertram[2] & Scot T. Martin[1,6]

Hygroscopic growth and cloud condensation nuclei activation are key processes for accurately modeling the climate impacts of organic particulate matter. Nevertheless, the microphysical mechanisms of these processes remain unresolved. Here we report complex thermodynamic behaviors, including humidity-dependent hygroscopicity, diameter-dependent cloud condensation nuclei activity, and liquid–liquid phase separation in the laboratory for biogenically derived secondary organic material representative of similar atmospheric organic particulate matter. These behaviors can be explained by the non-ideal mixing of water with hydrophobic and hydrophilic organic components. The non-ideality-driven liquid–liquid phase separation further enhances water uptake and induces lowered surface tension at high relative humidity, which result in a lower barrier to cloud condensation nuclei activation. By comparison, secondary organic material representing anthropogenic sources does not exhibit complex thermodynamic behavior. The combined results highlight the importance of detailed thermodynamic representations of the hygroscopicity and cloud condensation nuclei activity in models of the Earth's climate system.

---

[1] John A. Paulson School of Engineering and Applied Sciences, Harvard University, Cambridge, MA 02138, USA. [2] Department of Chemistry, University of British Columbia, Vancouver, BC V6T 1Z1, Canada. [3] Department of Earth and Environmental Sciences, Chonbuk National University, Jeollabuk-do 54896, Republic of Korea. [4] EWRE Division, Department of Civil Engineering, Indian Institute of Technology Madras, Chennai 600036, India. [5] College of Urban and Environmental Science, Peking University, 100871 Beijing, China. [6] Department of Earth and Planetary Sciences, Harvard University, Cambridge, MA 02138, USA. Correspondence and requests for materials should be addressed to S.T.M. (email: scot_martin@harvard.edu)

Organic compounds comprise a significant fraction of submicron, atmospheric particulate matter (PM)[1]. Organic PM is produced predominantly as secondary organic material (SOM), meaning by the in situ atmospheric oxidation of volatile organic compounds, emitted in a primary step by the Earth's biosphere[1,2]. Organic PM affects the Earth's energy budget, both directly by scattering and absorbing solar radiation and indirectly by serving as cloud condensation nuclei (CCN). The direct and indirect radiative effects of organic PM represent some of the largest sources of uncertainty for understanding of current and future climate change[3]. The interactions between organic PM and water vapor through hygroscopic growth at subsaturation conditions (i.e., <100% relative humidity (RH)) as well as through the activation as CCN at supersaturation conditions, strongly affect climate predictions and associated uncertainties[4].

Hygroscopic growth and CCN activation can be described by Köhler theory. The formulation takes into account the interplay between the Raoult solute effect of dissolved species and the Kelvin effect for small particles[5]. The full equations of Köhler theory have been simplified for use in atmospheric chemical models by employing a single hygroscopicity parameter $\kappa$ to represent the solute effect and assuming a surface tension of water[6]. An implied assumption of this $\kappa$-Köhler formulation is a negligible influence by surface-active organics at cloud activation. This $\kappa$-Köhler approach has successfully described many laboratory results and field observations related to hygroscopic growth for subsaturation and CCN activation for supersaturation conditions[7–10]. Even so, other laboratory and field studies suggest that the CCN activation of organic PM occurs at lower supersaturations than predicted based on $\kappa$ values derived for subsaturated condition[11–15].

Explaining the enhancement of CCN activity relative to predictions based on $\kappa$ values derived for subsaturated condition has been a focus of recent research where various mechanisms have been proposed[14–18]. One possibility is that surface-active organic solutes can lower the surface tension, leading to an enhancement of CCN activity. A laboratory study suggested the formation of a surface film of surface-active organics solutes for submicrometer particles of organic/inorganic mixtures at high RH[19]. The organic film and associated surface tension depression can explain the large droplet activation diameters for particles composed of SOM and ammonium sulfate[20]. In addition, surface tension depression was observed for cloud water collected from the ambient air[21], and the effect on CCN activation was modeled using a surface-bulk partitioning framework considering surface thermodynamics combined with the Köhler theory[22].

The enhancement of CCN activity relative to predictions based on $\kappa$ values derived for subsaturated condition might also be related to the presence of liquid–liquid phase separation (LLPS) for organic-containing particles at high RH[15,23–26]. Phase separation may occur in organic/inorganic mixtures, and the RH range depends on the oxidation state of organic components[26,27]. LLPS was also observed at high RH values (>95%) in SOM derived from α-pinene, limonene, and β-caryophyllene without inorganic inclusions[24,28]. LLPS is mainly driven by the non-ideality caused by mixing between water and solutes[29]. LLPS is associated with the formation of a water-rich phase containing water and hydrophilic species in the bulk, which can lead to an increase of water uptake for the particle of given diameter[24,25]. Likewise, LLPS is also associated with the formation of a hydrophobic, organic-rich phase on the droplet surface that reduces the surface tension[23]. When LLPS occurs at the high water activities relevant to CCN activation, both the corresponding increase in diameter and the reduced surface tension lowers the barrier of CCN activation[23].

An alternative mechanism to explain the enhanced CCN activity is the gradual dissolution of a sparingly soluble core as dilution occurs with water uptake (Supplementary Fig. 1 panel A). This mechanism has been described by a modified $\kappa$-Köhler formulation that incorporates the limited water solubility of different organic species[16,30–32]. This mechanism, however, seems to be unlikely for SOM because the presence of a large number of homogeneously mixed organic compounds in SOM can facilitate the formation of a stable liquid or amorphous state, even in the absence of water[33,34]. Another possible mechanism describes water uptake by surface water adsorption on highly viscous organic PM[14]. Surface adsorption would enhance water uptake at low RH, which is consistent with hygroscopicity data available from 50% to 90% RH[14]. This adsorptive mechanism, however, appears to be contradictory to recent measurements showing that water diffusion in typical-sized atmospheric particles can be slow but still occurs at timescale <1 s even at RH < 30%[35–37]. Although different explanations can reconcile and may support existing experimental data, whether these mechanisms are physically valid is not fully resolved. This uncertainty is in part because of the lack of a combined data set of high-quality hygroscopicity data at RH < 40% and of size-resolved CCN data at different supersaturation values. The hygroscopicity data at RH < 40% are valuable for elucidating water uptake mechanisms. Size-resolved CCN measurements provide information to model the effects of surface tension lowering.

In the present study, LLPS, hygroscopic growth (for RH values from 10% to 95%), and size-resolved CCN activation are investigated for laboratory-generated SOMs derived from different types of precursors representative of both biogenic and anthropogenic precursor sources. The data sets are interpreted in a solution-theory-based thermodynamic framework that considers the non-ideal mixing of water with hydrophobic and hydrophilic organic compounds, including the possibility of facilitating LLPS and lowering surface tension. The thermodynamic parameters are constrained by the hygroscopic growth measurements obtained for a wide RH range using a quartz crystal microbalance, and the model predictions are compared with observed LLPS behaviors and size-resolved CCN activity. The findings in the current study highlight the differences observed among different types of SOM. For SOMs representative of monoterpene-derived biogenic organic PM, the non-ideality and associated LLPS are observed, which can have an important role in determining hygroscopicity and CCN activity. By comparison, no LLPS is observed for SOMs representative of anthropogenic organic PM derived from aromatic and alkane precursors, and the associated hygroscopicity and CCN activity are well described by the $\kappa$-model.

## Results

**Biogenic SOM**. Submicron SOM particles were produced by the ozonolysis of α-pinene and limonene in an oxidation flow reactor. α-Pinene and limonene are abundant monoterpenes emitted as volatile organic compounds (VOC) from plants. Hygroscopic growth of the SOM thin films was measured by a quartz crystal microbalance (QCM) (see Methods). Humidity in a flow cell was switched between dry (<1% RH) and wet (10–95% RH) conditions. The response of the film mass to the RH change was continuously monitored. The mass-based hygroscopic growth factors, defined as the mass ratio of the film at an elevated RH to that at <1% RH, were obtained for α-pinene- and limonene-derived thin films (Fig. 1a). At low RH, SOMs can adopt amorphous semisolid- or solid-phase states[38–42]. Such highly viscous physical states may influence the hygroscopicity if water diffusion is sufficiently slow[38]. In the present study, kinetic limitation of in-particle water diffusion was accounted by measuring

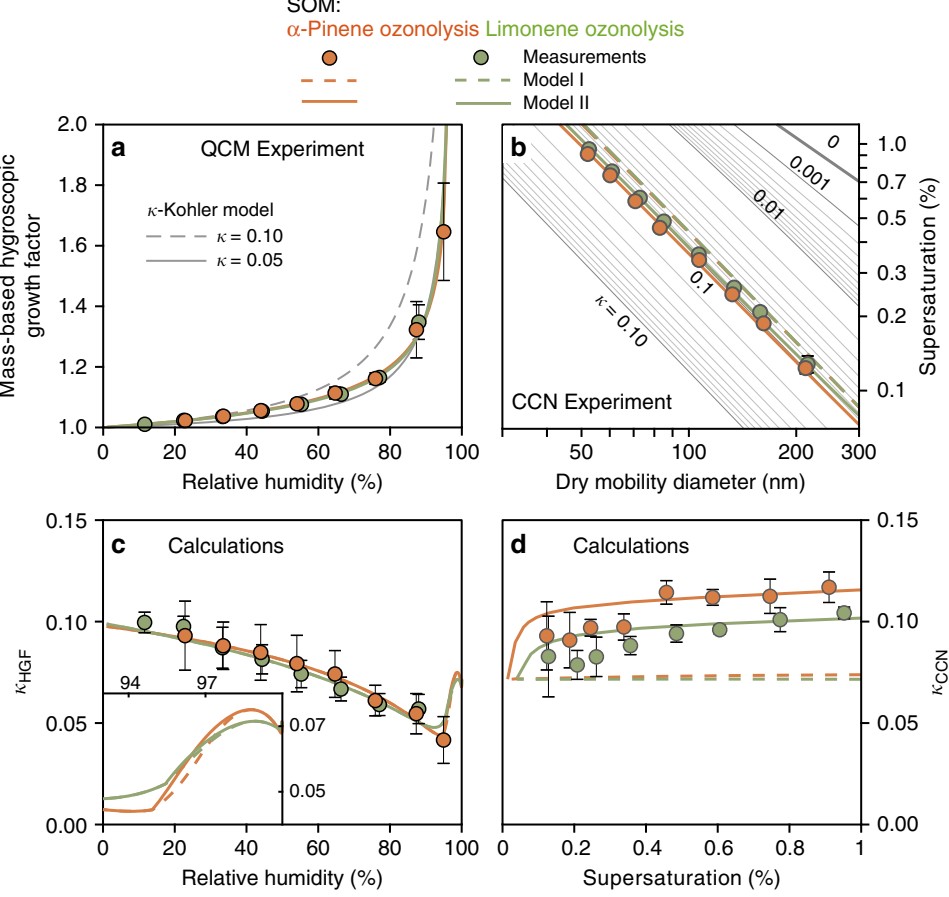

**Fig. 1** Hygroscopic growth and CCN activity of secondary organic material derived from ozonolysis of biogenic monoterpenes. **a** Mass-based hygroscopic growth factors $g_m$ measured by the quartz crystal microbalance for different organic films as a function of increasing relative humidity. **b** Critical supersaturation ratios $s_c$ (%) of CCN activation as a function of dry diameter $d_{m,dry}$ (nm). Gray lines in **a**, **b** show modeled results using constant $\kappa$ values. **c** Hygroscopicity parameters $\kappa_{HGF}$ calculated from the data shown in **a**. **d** Effective hygroscopicity parameter $\kappa_{CCN}$ calculated based on the size-resolved CCN measurements for the surface tension value of pure water (0.072 N m$^{-1}$). Colored dashed and solid lines in **a**–**d** represent results of Model I (single-phase model) and Model II (LLPS model), respectively (see main text). In **a**, dashed lines of Model I are hidden behind the solid lines of Model II because two models yield identical results in the absence of LLPS (RH < 95%). For clarity, model lines for $\kappa_{HGF}$ at high RH are shown in the inset of **c**

hygroscopicity after sufficient time for equilibration. Water uptake by the thin films was dominated by absorption. The mass of water taken up linearly increased with the mass of the film (Supplementary Fig. 2). Surface adsorption, characterized by the water uptake of the SiO$_2$ coated clean sensor, was subtracted as a baseline. Surface adsorption typically accounted for <10% of the total water uptake.

The film mass increased gradually from 10% to 95% RH due to uptake of water. Similar growth factor values were observed for organic films derived from α-pinene and limonene at a fixed RH value. The results of the $\kappa$-Köhler parameterization are represented by the gray lines in Fig. 1a. A single value of the hygroscopic parameter $\kappa$ failed to accurately capture the observed growth curves when considering the whole RH range. Figure 1c illustrates the RH-dependent values of hygroscopicity parameter $\kappa_{HGF}$, calculated based on the measurements represented in Fig. 1a. $\kappa_{HGF}$ varied from 0.1 for RH < 20% to <0.05 at >90% RH, indicating that water uptake of the SOMs deviated significantly from the behavior of ideal solutions. The decreasing trend of $\kappa_{HGF}$ with RH cannot be explained by the modified $\kappa$-Köhler formulation that assumes limited water solubility of organic compounds. This formulation predicts a monotonic increase of $\kappa$ values for an increasing RH because the sparingly soluble organic

compounds gradually dissolve into the aqueous phase and thus contribute to the hygroscopicity[14–16,30,31]. A similar decreasing trend of $\kappa$ observed for biogenic SOM, as shown in Fig. 1c, has been previously interpreted as surface adsorption[14,15]. The results presented herein indicate that the decreasing trend persists even after the correction for surface adsorption.

Size-resolved CCN measurements were conducted for supersaturation ranging from 0.15% to 1%. For each supersaturation value $s_c$, a critical dry mobility diameter $d_{m,dry}$ of CCN activation was determined. Results for α-pinene and limonene-derived SOM particles are illustrated in Fig. 1b. For comparison, curves of modeled $s_c$ vs. $d_{m,dry}$ for a series of $\kappa$ values were computed by the single parameter $\kappa$-Köhler model based on a surface tension of water ($\sigma_{s/a} = 0.072$ J m$^{-2}$), as represented by the gray lines in Fig. 1b. The measured $s_c$–$d_{m,dry}$ data for α-pinene- and limonene-derived SOMs correspond to the model line of $\kappa = 0.1$, implying that CCN activity was significantly higher than the hygroscopicity measured at 85–90% RH ($\kappa$ of 0.05, Fig. 1c). At a fixed supersaturation, a shift in $\kappa$ value from 0.05 to 0.1 can lower the critical dry diameter by 20%, meaning that a higher number of particles can be activated as CCN in the context of cloud microphysics[43]. Effective $\kappa_{CCN}$ values were calculated assuming a surface

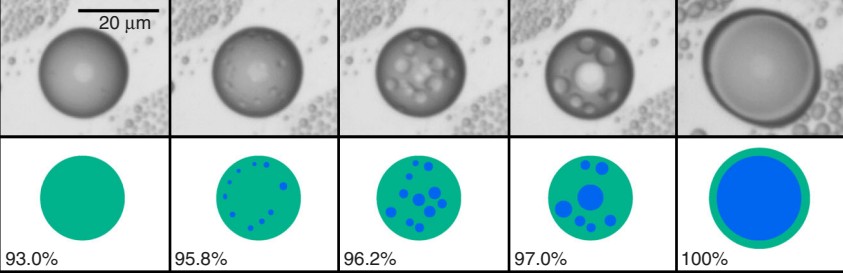

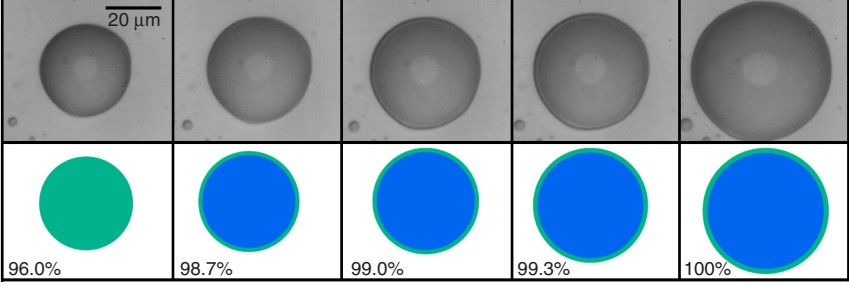

**Fig. 2** Optical microscopic images show the presence of LLPS at high relative humidity for supermicron SOM particles derived from α-pinene and limonene ozonolysis. Illustrations of the images are shown for clarity. Images for **a** α-pinene-derived SOM and **b** limonene-derived SOM were taken from previous experiments described in ref. [24] and ref. [28], respectively. Images are shown for an increasing RH. Small inclusions in **a** show the formation of a second liquid phase. The light gray circles at the center of the particles are an optical effect of the microscope and therefore are excluded from the illustrations

tension of pure water. Figure 1d shows that $\kappa_{CCN}$ increased with an increasing $s_c$ (i.e., a decreasing $d_{m,dry}$).

Several previous studies interpreted the enhanced CCN activity relative to hygroscopicity for subsaturation results from gradual dissolution of sparsely soluble organic compounds, as expected by the solubility-based formulation[14,16]. This interpretation, however, in part appears to be contradictory with the size-resolved CCN observation shown in Fig. 1d. For particles with an identical chemical composition, Köhler theory predicts that the dilution ratio at the activation point decreases with an increasing $s_c$ (i.e., a decreasing $d_{m,dry}$). In the solubility-based formulation, a decreasing dilution ratio results in less organic solute dissolved in the aqueous phase, thus decreasing the effective $\kappa$ value. This prediction is inconsistent with the size-resolved CCN measurements which show that the effective $\kappa$ value increases for an increasing $s_c$ (Fig. 1d).

Microscope images of supermicron SOM particles generated from the oxidation of alpha-pinene and limonene are represented in Fig. 2a, b. These images are adapted from previously conducted experiments[24,28], and the measurements are described in the Methods section. Across figure columns, RH was stepwise increased. α-Pinene-derived SOM formed a uniform single phase for RH < 95% (Fig. 2a). Water and organic compounds were miscible with each other over this range of RH values[44]. For RH < 95%, the mass fraction of organic compounds dominated relative to water (Fig. 1a). The abundant hydrophilic organic molecules with intermediate polarities can possibly bridge the miscibility gap between non-polar hydrophobic molecules and polar water molecule, thereby facilitating the formation of a single phase. Above 95% RH, water-rich aqueous inclusions formed inside the supermicron particle. As RH was further increased, the mass fraction of water dominated relative to organic compounds (Fig. 1a), and the particle reached an equilibrium state consisting of a hydrophobic organic-rich shell and an aqueous core dominated by water and

hydrophilic organics (Fig. 2a). LLPS was also observed for limonene-derived SOM. Figure 2b shows the formation of a thin surface film above 98% RH. The phase separation persisted until the organic-rich phase completely dissolved as water activity approached unity.

**Modeling approach.** To explain the complex behaviors of water uptake and CCN activation observed for the biogenic SOMs, we developed a Flory–Huggins–Köhler model framework for SOM[25]. The model framework accounted for the interplay of non-ideal solution, LLPS, and surface activity. The solution-theory-based thermodynamic framework used here is similar to the model framework used recently to successfully explain the observed high CCN number concentrations of organic/inorganic mixed particles in marine air masses and to establish the importance of LLPS at the point of activation using a field data set[23]. A major difference is that the current framework does not require detailed input of molecular properties, such as functional groups and molecular weights of organic compounds[23,29], which remains poorly constrained for atmospheric organic PM.

The Flory–Huggins solution theory is a thermodynamic model that takes an account of the dissimilarity in molecular sizes between small molecule solvents and large molecule solutes. The non-ideal mixing can be captured by including an interaction parameter $\chi$ that describes the enthalpy of mixing between organics and water (Supplementary Methods). LLPS and phase equilibrium between two liquid phases can be calculated by minimizing Gibbs free energy following the method described in Zuend et al.[29]. Partitioning of water and organic components between shell and core phases is calculated for the changing water activity. The presence of LLPS is indicated by the phase equilibrium of two solutions of distinct chemical compositions. When LLPS occurs, surface tension is calculated using a semiempirical approach considering complete or partial coverage

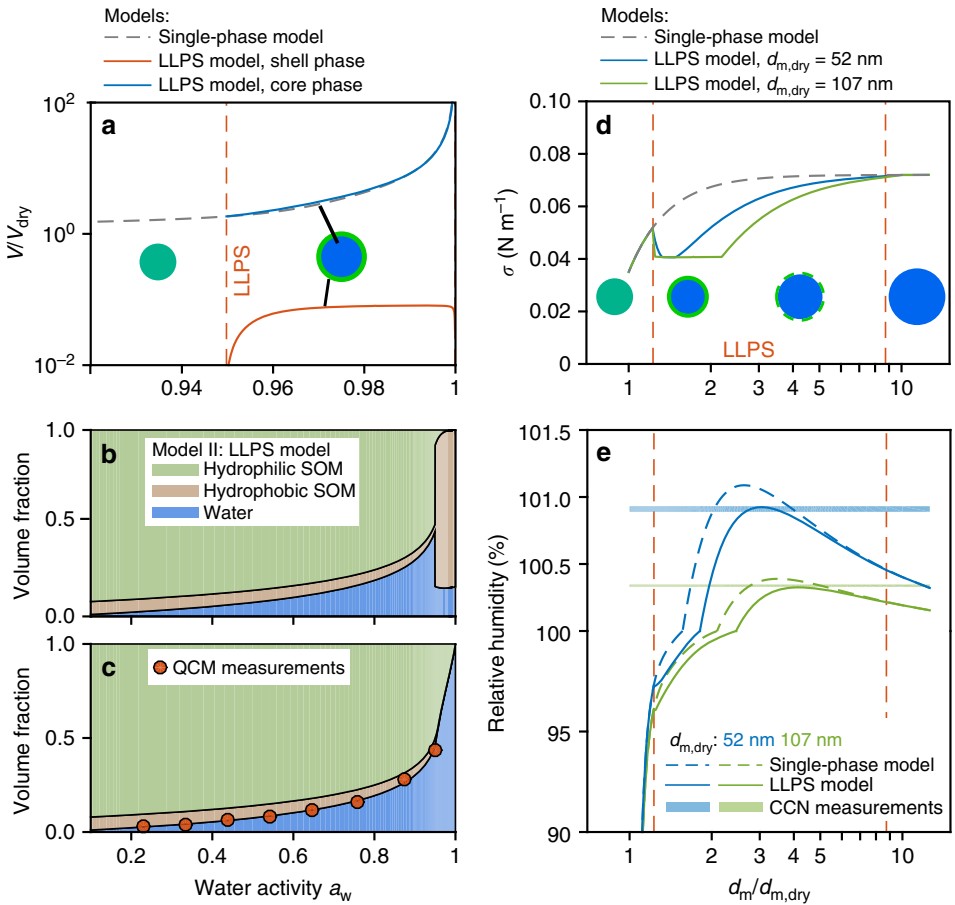

**Fig. 3** Thermodynamic model predictions for an α-pinene-derived SOM particle. **a** Volume $V$ of single and separated phases across a water activity range of 0.92–1. The dashed gray line shows the single-phase case predicted by Model I. Orange and blue lines show the shell and core phases, respectively, predicted by Model II when LLPS is present. Values are normalized by the volume $V_{dry}$ of organic material when dry. The red dashed vertical line marks the predicted onset conditions of LLPS. The $V/V_{dry}$ value of the shell phase drops off when water activity approaches unity because of the mixing of two phases into a single phase. Predicted volume fractions of hydrophilic, hydrophobic organic components, and water are shown in **b** for the surface and in **c** for the core of the particle. The discontinuity of hydrophobic SOM volume fraction shown in **b** represent onset of the LLPS. The volume fractions of water calculated based on the mass-based QCM measurements of hygroscopic growth are shown in **c** for comparison. **d** Predicted surface tension coefficient $\sigma$ vs. the diameter growth factor $d_m/d_{m,dry}$. Predictions are made for dry diameters of 52 nm (blue) and 107 nm (green). Cartoons in **a**, **d** show the possible morphology of the particle. Blue and green colors represent water and organics, respectively. **e** Köhler curves of relative humidity vs. diameter growth factor predicted based on the surface tension shown in **d**. Measured supersaturations for 52 and 107 nm diameter particles are marked as horizontal lines for comparison. The thicknesses of the lines are scaled by the measurements' uncertainties

of surface active organic monolayer, similar to the approaches used in Ovadnevaite et al.[23]. and Ruehl et al.[20].

Two model configurations with different levels of complexity in the treatment of compositions were deployed in this study. The aim is to provide a multi-component framework with minimum complexity in organic composition, which possibly can be obtained using experimental data and that can still adequately describe the observed complex behaviors of hygroscopic growth, LLPS, and CCN activity.

The first model (Model I: single-phase model) described an effective binary mixture of SOM and water (Supplementary Methods). The $\chi$ values were derived experimentally from the hygroscopic growth measurements by the QCM for α-pinene-derived SOM (Supplementary Methods). The values of $\chi$ were fitted as a function of organic volume fraction and further extrapolated to diluted conditions that are relevant to cloud activation (Supplementary Fig. 3). By accounting for the non-ideality of mixing between water and organic molecules, this model predicted a decrease of $\kappa$ at water activity <0.95 and an increase of $\kappa$ at water activity >0.95, which in part closed the gap

between the hygroscopic growth and CCN measurements. Even so, by simplifying the SOM as a single compound, the binary model always predicted that SOM and water form a single phase at any mixing ratio (Fig. 3a), meaning that the observed LLPS was not captured. The pure-component SOM was treated as having a surface tension value lower than that of water. In the absence of LLPS, however, the surface concentration of organic solute was assumed to be equal to that of the bulk solution. The SOM mainly contributed to the CCN activity by the Raoult solute effect. The calculated surface tension was close to that of pure water at conditions relevant to CCN activation (Fig. 3d) due to the dilution of organics. As a result, the predicted $\kappa_{CCN}$ values were underestimated by 10–40%, and the observed $s_c$-dependent behavior of $\kappa_{CCN}$ was not captured (Fig. 1d). Sensitivity tests revealed that the observed $s_c$-dependent behavior of $\kappa_{CCN}$ can only be explained by assuming a surface tension lower than that of pure water, indicating a substantial surfactant effect of the organic compounds.

The second model (Model II: LLPS model) considered a ternary system consisting of hydrophilic/hydrophobic fractions of

SOM and water (Supplementary Methods). The hydrophobic fraction of SOM was assumed to comprise a small fraction of total SOM (5–7% by mass; Supplementary Table 1), but the value of interaction parameter of water/hydrophobic compounds was relatively large, meaning a large positive enthalpy of mixing (Supplementary Table 1). When the hydrophobic component was mixed with water, the large positive enthalpy of mixing could result in a miscibility gap at a wide range of mixing ratios, consistent with the hydrophobic properties of these compounds. The hydrophilic fraction of SOM was treated as having a relatively small positive interaction parameter with water, indicating a small mixing enthalpy with water. It was miscible with water at any mixing ratios. Further description is in Supplementary Methods. For water activity >0.95, the model predicted that a two-phase core–shell configuration had a lower Gibbs free energy than a single phase, indicating that LLPS was thermodynamically favorable. This predicted LLPS behavior is consistent with the observations shown in Fig. 2a, b (Supplementary Table 1). Once formed, the volume of the shell phase, as calculated in the model by assuming phase equilibrium, was insensitive to water activity, while the volume of the core grew by orders of magnitude at higher water activity (Fig. 3a). As a result, the shell phase might drop below full surface coverage of the droplet when its thickness was less than a minimum thickness $\delta$, describing a monolayer. The treatment of partial coverage was also consistent with an organic film model used in recent studies[20,23]. The shell phase was predicted to be organic rich, consisting mainly of hydrophobic SOM and a small fraction of water (Fig. 3b). Conversely, the core phase was water rich, composed of water and hydrophilic SOM (Fig. 3c).

These results indicate that chemical composition of the particle surface region can differ significantly from that of the interior region in the presence of LLPS. One implication is that the organic-rich shell can lower surface tension of the particle/air interface at high RH values, thereby facilitating cloud activation. This LLPS effect of lowering surface tension is emphasized by comparison of the predicted surface tension ($\sigma$) values of Model II (LLPS model) vs. Model I (single-phase model), as shown in Fig. 3d. The predicted LLPS state by Model II is denoted by the region between the two vertical dashed lines. In the LLPS region, the predicted $\sigma$ values from Model II are up to 40% lower than those of Model I. For a growing droplet with an increasing diameter growth factor $d_m/d_{m,dry}$, the $\sigma$ value predicted by Model II decreases in abrupt fashion at the transition from a single phase to LLPS. The $\sigma$ value increases again at a point where the minimum thickness $\delta$ is reached. This point occurs at a lower $d_m/d_{m,dry}$ value for a smaller dry diameter. The LLPS state is maintained until the particle reaches a large dilution ratio ($d_m/d_{m,dry}$ approaching 9), corresponding to a high $a_w$ value of 0.9999 (Supplementary Table 1). Above this point, the particle becomes one phase, and the $\sigma$ value is effectively that of pure water. In comparison, the $\sigma$ value predicted by Model I monotonically increases along with dilution, and the $\sigma$ value becomes close to that of pure water when the $d_m/d_{m,dry}$ value is >2.0.

The Köhler curves calculated for both the models are illustrated in Fig. 3e. The CCN activation points for typical-sized atmospheric particles, as shown by the maximum of the Köhler curves, are within the LLPS region predicted by Model II. In line with the foregoing understanding, Model II predicts a lower critical supersaturation of CCN activation than that predicted by Model I. Both are calculated for a fixed particle dry diameter. For comparison, the measured critical supersaturation values are illustrated for two dry diameters as the horizontal lines in Fig. 3e. Predictions by Model II agree well with the size-resolved CCN measurements, while predictions by Model I overestimate the critical supersaturation values.

**Anthropogenic SOM**. Microscopic morphology, hygroscopic growth, and CCN activation measurements were also performed for SOM derived from the photooxidation of the anthropogenic precursors toluene and dodecane (Figs. 4, 5). Toluene is an abundant aromatic hydrocarbon, and dodecane is a representative alkane compound in the atmosphere. Both compounds are primarily emitted from anthropogenic sources. In contrast to the SOM derived from biogenic α-pinene and limonene, microscopic images show that toluene- and dodecane-derived SOMs do not form a LLPS state (Fig. 5a, b). A single phase is sustained over the entire investigated RH range. Figure 4a, b show that the single-parameter $\kappa$ model fits reasonably well both the hygroscopic growth curves and the CCN activation data. The calculated $\kappa$ values do not significantly vary with RH or $s_c$ (Fig. 4c, d). The SOM derived from toluene has a higher $\kappa$ value than does the SOM derived from dodecane (i.e., 0.19 vs. 0.12). The higher $\kappa$ value is associated with a higher oxygen-to-carbon (O:C) ratio (Supplementary Table 2). The relationship with O:C is consistent with the findings of previous studies[1,10]. These results suggest that the widely used $\kappa$-Köhler parameterization is applicable for the investigated anthropogenically derived SOMs, for which LLPS is absent.

## Discussion

The experimental and modeling results presented in this work elucidate the mechanisms of organic PM water uptake. The high sensitivity of the QCM method substantially improves the precision and accuracy of water uptake measurements in the low RH region. By excluding the influence of water adsorption, the observed high hygroscopicity and gradual water uptake at low RH reveals that water uptake of SOM is not solubility limited. This result is expected in multi-component systems where water and different organic compounds can be miscible with each other and form stable liquids or amorphous states[33]. This inference is consistent with the direct microscopic measurements showing that SOM formed a single-phase morphology at low RH in the absence of an undissolved insoluble core. The implication is that hygroscopicity and CCN activity for amorphous organic substances may not be directly determined by their solubility. Other thermodynamic factors, such as the polarity and the mixing enthalpy with water and other organic molecules, can be important. Previous solubility-based model frameworks may not adequately describe the water uptake and CCN activation behaviors of amorphous organic PM. To reconcile the discrepancy between the measurements and the solubility-based model predictions, previous studies have hypothesized that surface adsorption can be a predominant mechanism for water uptake of highly viscous SOMs (e.g., as characterized by particle rebound)[14,15]. Direct measurements made for organic thin films in this study, however, show that water uptake depended on the volume of the organic material rather than the surface area (Supplementary Fig. 2), indicating that water uptake was dominated by absorption and that surface adsorption was negligible.

The QCM method also provides time-resolved measurements and possible kinetic effect for hygroscopic growth was considered. Water uptake was determined after equilibrium, and the complex behaviors of water uptake reported herein should be interpreted as a state of thermodynamic equilibrium rather than a kinetic effect. For atmospheric particles, the equilibration time with surrounding water vapor largely depends on the RH, temperature, particle size, and SOM type[35,36,45]. The equilibration times for water uptake of typical-sized atmospheric particles are estimated to be <1 s for temperature and RH conditions relevant to the atmospheric boundary layer, even though some particles can have semisolid or solid physical states[35]. The implication is that the

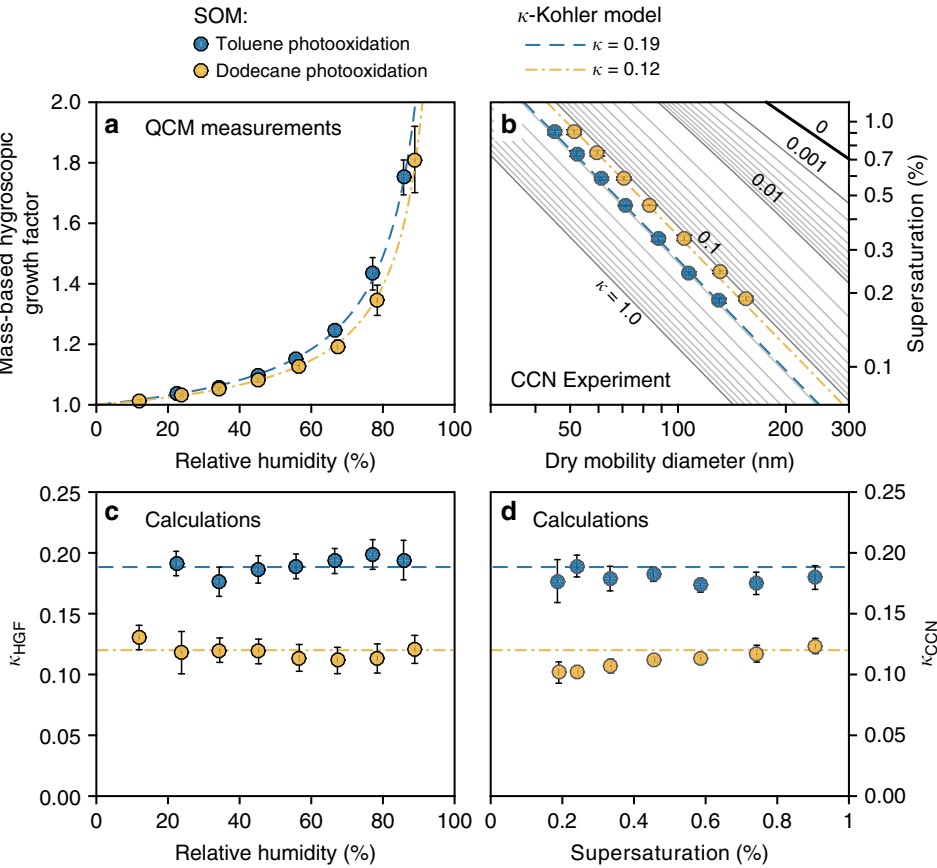

**Fig. 4** Hygroscopic growth and CCN activity of secondary organic material derived from the photooxidation of the anthropogenic compounds toluene and dodecane. **a** Mass-based hygroscopic growth factors measured as a function of relative humidity. Dashed lines show modeled results using constant values of $\kappa$ based on the experimental data. **b** $s_c$–$d_{m,dry}$ data measured for SOM particles derived from photooxidation of toluene and dodecane. Dashed lines show the predicted CCN activity based on the average values of $\kappa$ measured at subsaturation. **c** Hygroscopicity parameter $\kappa_{HGF}$ calculated from results shown in **a**. **d** Effective hygroscopicity parameter $\kappa_{CCN}$ calculated based on the size-resolved CCN measurements for the surface tension of water

equilibrium with water vapor within an hourly time step in the chemical transport models should be an accurate assumption.

The simplified ternary mixture model considering water, hydrophobic species, and hydrophilic species can bridge the macroscopically observed discrepancy between CCN activity and hygroscopicity with microscopically observed LLPS. The results highlight the important role of a small fraction of hydrophobic organic compounds in water uptake and CCN activation. The mixing enthalpy between water and hydrophobic organic component can give rise to non-ideal mixing behaviors deviating from the Raoult's law, as represented by the RH-dependent hygroscopicity observed for the biogenically derived SOMs. Thermodynamic predictions in this work reveal that such enthalpic term can lead to LLPS as observed for the same type of SOM at high RH conditions relevant to CCN activation. Modeling results also suggest that the hydrophobic organics preferably partition to the shell phase when LLPS occurs, and consequently reduce the surface tension enhancing their CCN activation potential. Surface tension lowering is experimentally supported by the observed size-dependent CCN activity in this work for biogenically derived SOM particles. This contribution of hydrophobic compounds to CCN activity is not considered in the conventional solubility-based model framework, in which the hydrophobic component is treated as an insoluble core and thus does not contribute to CCN activity.

As a caveat, the model framework employed herein only considered interactions between organic solutes and water. This

treatment is appropriate for the present work because the generated SOM was free of inorganic inclusions. Particles in the atmosphere, however, are usually internal mixtures of organic and inorganic species. In this case, the inorganic species can cause salting out, resulting in increased LLPS[46]. Ionic interactions of electrolyte with water and non-electrolytes should be included in the model in future studies.

For inorganic/organic mixtures, LLPS more readily occurs when the O:C ratio of the organic component is low[27,47].The fraction of low polarity compounds is a key factor in determining whether the LLPS occurs[24], and the O:C ratio serves as a proxy for polarity of a molecule[48]. Recent literature suggests that including the surface-tension-lowering effect induced by the interfacial organics can reconcile the measured CCN number concentration of inorganic–organic mixed aerosol particles in a marine airmass[23] as well as the observed wet diameter of CCN activation for SOM with inorganic seeds[20], both highlighting the important role of LLPS in cloud droplet activation.

The results presented herein also highlight the differences between SOMs representative of biogenic compared to anthropogenic sources. Strong non-ideality, as illustrated by the presence of LLPS, along with the RH dependence and the discontinuity below and above water vapor saturation in apparent hygroscopicity, was observed for α-pinene- and limonene-derived SOMs representing biogenic precursor sources. It was not observed for toluene- and dodecane-derived SOMs representing anthropogenic precursor sources. This study investigated four

**a**  SOM derived from toluene photooxidation (OH)

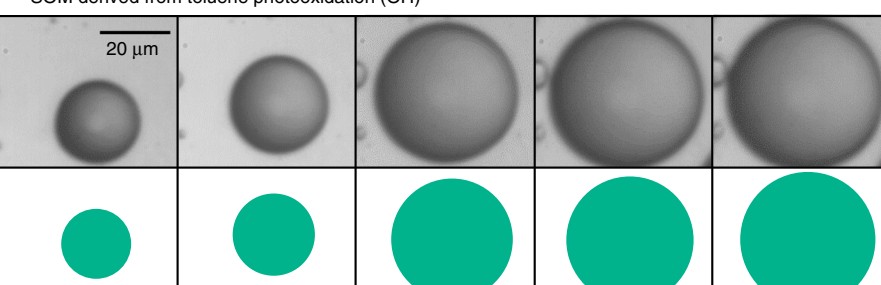

**b**  SOM derived from dodecane photooxidation (OH)

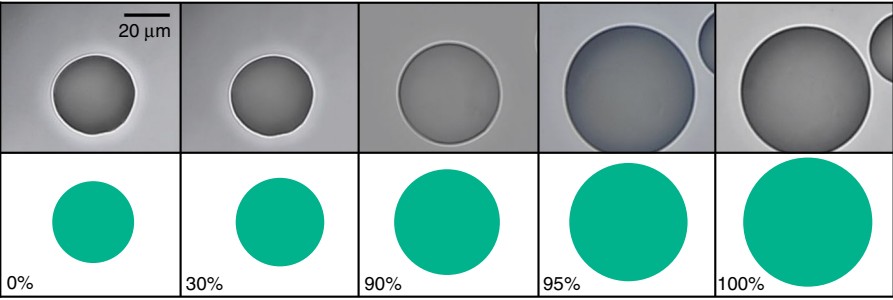

**Fig. 5** Optical microscope images show the absence of LLPS for the entire relative humidity range. Illustrations of the images are shown for clarity. SOM particles were derived from **a** toluene photooxidation and **b** dodecane photooxidation. Images for toluene-derived SOM were taken from previous experiments[28]

SOMs, and the properties of SOMs derived from other biogenic and anthropogenic precursor sources should be investigated to examine the further generalization of the results. By implication, the studied biogenically derived SOMs contain significant amounts of low polarity organic compounds[49]. Future laboratory and field studies are needed for improvement of molecular characterization and better understanding of the physical and chemical properties of these compounds.

## Methods

**Production of SOM**. Different types of SOM were produced by oxidizing gaseous precursor in an oxidation flow reactor (OFR) without using inorganic seed particles[42]. α-Pinene- and limonene-derived SOMs were produced from dark ozonolysis. Toluene- and dodecane-derived SOMs were produced in the OFR by photooxidation of the precursors primarily by hydroxyl radicals. Non-OH pathways, such as the photolysis of VOC precursors, were estimated to account for <1% of precursor loss[50]. The produced particle populations were characterized using a scanning mobility particle sizer (SMPS; TSI Inc.) and a high-resolution time-of-flight aerosol mass spectrometer (AMS; Aerodyne Research Inc.). Mass concentration and elemental ratios were characterized by the SMPS and AMS, respectively. The results are listed in Supplementary Table 2.

**Optical microscopy**. Detailed experimental description of the optical microscopy measurements can be found in the previous publications[24,28]. In brief, sub-micrometer SOM particles produced in the OFR were collected onto a hydrophobic substrate using a single-stage particle impactor. After sufficient deposition, large particles with size of 20–80 μm formed on the substrate from coagulation of submicrometer particles upon impaction. Optical images were recorded by an optical microscope (Zeiss Axiotech, 50× objective and Olympus BX43, 40× objective). Images presented herein were obtained for increasing RH.

**Quartz crystal microbalance**. Thin films of SOM were grown by electrostatic precipitation of aerosol particles onto SiO₂-coated QCM crystals (Q-sense QSX 303) using a Nanometer Aerosol Sampler (TSI 3089)[42,51]. Film thicknesses was on the order of 100 nm. The SOM-laden QCM crystals were mounted into a temperature- and humidity-controlled flow module. Different relative humidities, as continuously monitored by an RH sensor (Rotronic, HydroClip 2), were achieved by changing the mixing ratio of dry and humidified nitrogen flows using two mass flow controllers (MKS M100b). The total flow rate was kept at 20 cm³ min⁻¹. Film

mass was continuously monitored by the QCM (Q-sense E4). A schematic diagram of the QCM measurement can be found in a previous publication[42]. Dry mass (<1% RH) was measured before and after the measurement for each elevated RH, and possible evaporation of dry organic material was accounted for. The mass sensitivity of the QCM was <1 ng cm⁻², corresponding to one tenth of a single-molecule-layer water. This sensitivity was sufficient for accurate detection of water uptake for RH > 10%. The QCM method was validated by experiments using amorphous sucrose thin films, giving good agreement with literature (Supplementary Fig. 4). Hygroscopicity parameter $\kappa_{HGF}$ was calculated from the QCM measurements[6], using the material density calculated based on the elemental ratios[52] (Supplementary Table 2).

**CCN counter**. Polydisperse SOM particles produced from the OFR were size-selected by a differential mobility analyzer (TSI 3081). The particles were dried using a Nafion dryer (Perma Pure Inc.), and the RH prior to entering the CCN counter was kept below 20%, as confirmed by an RH probe (Omega, RH-USB). The CCN activation ratio was determined as a function of particle dry mobility diameter from simultaneous measurements of a continuous flow CCN counter (DMT CCNC) and a condensation particle counter (TSI 3772). Supersaturation in the CCNC varied between 0.15% and 1%. Calibration was based on the activation of ammonium sulfate particles. Critical dry diameter $d_{m,dry}$ for each supersaturation value was determined from fitting of the activation curve to a Gauss error function. The doubly charged particle fraction was subtracted prior to data fitting. The detailed protocol of size-resolved CCN calibration and measurement followed that described in Rose et al.[53].

## Data availability

The experimental data and model code that support the findings of this study are available from the corresponding author upon reasonable request.

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

## Acknowledgements

The authors acknowledge Jianhuai Ye, Claudia Marcolli, and Andreas Zuend for fruitful discussions. We thank Onye Ahanotu for assistance with the quartz crystal microbalance (QCM) instrument and Wen-Chien Lee and Mikinori Kuwata for assistance with the measurements of SOM soluble fraction. The QCM experiments were performed at the Material Characterization Core of the Wyss Institute for Biologically Inspired Engineering at Harvard University. This research was funded by the Atmospheric System Research Program of the Office of Science of the Department of Energy (DE-SC0012792) and by the Geosciences Directorate of the National Science Foundation (AGS-1249565).

M.S. acknowledges support from a National Research Foundation of Korea (NRF) grant funded by the Korean government (MSIP) (2016R1C1B1009243). S.S.G. was a recipient of the Fulbright Fellowship and is thankful to the office of International and Alumni Relation, IIT Madras, for partial travel funding.

## Author contributions

P.L., S.S.G., A.K.B., and S.T.M. designed the research project, interpreted all results, and contributed to writing. P.L., T.Z., S.S.G., Y.H., and J.C.A. performed QCM and CCN measurements. M.S. and S.H. carried out optical microscopy experiments. P.L., Z.G., and Y.M.Q. performed AMS analyses. P.L. developed the thermodynamic model.

## Additional information

**Competing interests:** The authors declare no competing interests.

