## [Peer Review File · Nature Communications]

Reviewer #1 (Remarks to the Author):

The manuscript entitled, "Resolving the Mechanisms of Hygroscopic Growth. . ." by Liu et al., focuses on resolving the long standing question of why organic aerosols exhibit much greater apparent CCN activity under supersaturated conditions than would be predicted by measurements of growth factors conducted under subsaturated ($RH < 100\%$) environments. The authors present new measurements (and images) over a wide range of sub and supersaturated states for SOM of biogenic and anthropogenic origin. They also analyze the data using two models based upon the Flory Huggins Thermodynamic Model. The manuscript approaches an important problem for climate and atmospheric chemistry using new data and measurements. The manuscript is well written and the conclusions generally follow from the authors' analysis. I recommend the manuscript for publication when the authors have addressed the following in their revised manuscript.

1. The issue of LLPS has been discussed by numerous authors and only recently been applied in a robust way to CCN. For example in Ref 21 (Ovadnevaite et al., 2017). The authors never really clarify how their modeling approach is different, the same or complimentary to this prior work. This work, which is only vaguely referred to, established the importance of LLPS at the point of activation (albeit) using a field data set. The authors should make it very clear how their modeling work connects with Ovadnevaite et al. In other words, is the model entirely new? or is this paper mainly about presenting new data to validate an already established modeling approach.

2. Regarding surface tension (σ). The authors state in the SI that "In the present work, a σ value of 0.035 N m^{-1} was used for both hydrophobic and hydrophilic organic components. The effective surface tension of a liquid phase was calculated as the volume-weighted mean of σ values of the components." It is not clear to me why 0.035 was selected? How sensitive are the results to this number, is this a fitting parameter? I would like to see some sensitivity analysis to better describe this assumption.

Also why was volume weighted σ values used instead of surface weighted? Volume weighting can substantially under account for surface active species (i.e. surfactants).

3. Line 152-158: Here the authors rule out the gradual dissolution of sparingly soluble compounds as a way to reconcile sub and supersaturated κ . They point to Fig. 1F as evidence. Given the importance of this conclusion, the authors should expand this discussion giving the reader a clear idea of the logic in addition to pointing to the figure.

4. Model 1 (lines 184). It is not clear what the implications are for treating SOM as a surfactant? Does Model 1 allow SOM to lower water activity (or just surface tension). How is surface partitioning computed in this case. This should be clarified.

5. Lines 214-220 The authors say the volume of the shell phase was insensitive to changes in water activity (unlike the core). Is this an observation or a constraint in the model? In other words, does the model allow for water uptake into the shell and re-partitioning of material from the shell to core or core to shell as water activity changes? The authors should clarify this. If this model doesn't allow for dynamic re-partitioning between shell and core, the authors should defend this model simplification.

6. The Model (three components water, hydrophobic, and water soluble species) is clearly a simplification for the complexity of SOM, which in principle is ok and necessary. Some discussion in the manuscript of this simplification is needed with appropriate caveats. For example, what could be the role that small populations of extremely surface active species play and could these alter the main conclusions of the paper?

7. Lines 64-67 The role of larger than expected droplet diameters has been observed in both supersaturated regimes (Ref 20) but also by Ruehl et al. in sub saturated conditions where the role of monolayer structure and associated changes in surface tension was explicitly investigated (Journal of Physical Chemistry A 118 (22), 3952-3966). It seems appropriate to include this reference in this part of the discussion.

Reviewer #2 (Remarks to the Author):

Review of Liu et al., Resolving the Mechanism of Hygroscopic Growth and Cloud Condensation Nucleii Activity for Organic Particulate Matter

This paper claims that biogenically derived SOM exhibit non-ideal behavior in the form of LLPS, RH dependent hygroscopicity and discontinuous hygroscopicity at the point of water vapor saturation. It claims that this behavior is caused by non-ideal mixing of water with hydrophobic and hydrophilic organic components of the biogenically derived SOM, which are not present in anthropogenically derived SOM (which exhibit ideal behavior).

The non-ideal behavior of biogenically derived SOM has been observed by others, as pointed out on line 63 of the paper, as has LLPS in these specific SOMs (see figures 1 and 2). The novelty of this paper lies in obtaining size resolved CCN measurements and hygroscopicity data below 40% RH, and using these to parameterize and test a model to explain the observed behavior.

This paper will be of interest to others in the field as CCN activity is one of the larger uncertainties on aerosol-cloud-interactions and their effect on climate, and improved K values are needed to address this.

This paper challenges other ideas that have been proposed to explain enhanced CCN activity relative to predictions based on K values and provides a model/theoretical framework that could be tested on other SOM as a new way to understand CCN activity of SOM.

Convincing evidence that the proposed model, that considers polarity and mixing enthalpy with water and other organic molecules describes the observed behavior better than a model which describes an effective binary mixture of SOM and water, treating SOM as a surfactant.

While testing of this model on further biogenically and anthropogenically derived SOM would clearly be beneficial in determining its general utility, I do not consider that necessary for the publication of this work, provided this is properly commented on in the manuscript. Current lines 320-321 indicate the need of 'improvement' and 'better understanding of the physical and chemical properties of these compounds at a molecular level'. This falls short of a clear statement of the limitations of using only 4 compounds in this study and lacks clarity regarding what "improvements" the author considers necessary.

The manuscript lack clarity, mostly in how the figures are described and presented and in the structure of the figures themselves. The first figure referred to is figure 1C, whereas it is customary to order figures by order of discussion in the main text. The order problem comes up again for figure 3. It is perhaps unclear to me panels a and b of figures 1 and 3 need to be grouped with panels C-F, and navigation of the paper might be improved by putting these as separate figures. The legend for figure 2 panels C-F is not well placed. I found myself searching for it when looking first at panel C. It is unclear whether the dashed orange and green lines representing model 1 are just hidden behind the solid lines of model 2 in panels C and D, or not included, or indeed if these lines are the model or just a fit. Figure 2 panel a would benefit from an explanation of the drop off/increase in V/V_{dry} for the shell and core phases respectively as the water activity approaches unity. Similarly, the discontinuity in the hydrophobic SOM volume fraction in panel B at high water activity should be explained in the figure caption. The QCM measurements in panel C of figure 2 are not explained or referred to in the figure caption, and ought to be. The cartoons of particle phases in figure 2 panels A and D need reference to and explaining in the caption.

Clarity is also impeded by clumsy sentence structure of missing clauses, including the following:

In the abstract, line 32, the description of the behavior as "complex", especially as this complexity has already been remarked upon in line 30 is redundant, and I suggest the word be omitted.

Line 55 would read better with "the negligible influence" rather than "a negligible influence".

Line 261: “are valuable for elucidating’ could be just “elucidate”

Line 263: “improved the capability of water uptake measurements” – does this mean makes the measurements possible? Or more accurate? Or more precise? It could be better explained.

Line 320: “future laboratory and field studies are needed for improvement ... of the physical and chemical properties of these compounds”, needs to be better explained. Improvement in what sense?

The abstract does not currently provide a clear enough overview of the manuscript because of a failure to adequately explain some of the concepts introduced:

Line 28 states that the paper presents results of LLPS, hygroscopic growth and size resolved CCN activation studies, but a clear explanation of why LLPS is relevant for understanding “the mechanisms of hygroscopic growth and CCN activation” is both warranted and currently lacking.

Line 35 – it needs to be more clearly stated that it is alternative mechanism to explain humidity dependent hygroscopicity and size-dependent CCN activity that are being ruled out .

Lines 35-36, “For anthropogenically-derived organic PM, mechanistically different processes of CCN activation were observed because of the absence of LLPS” is ambiguous. What these processes are different to should at least be clearly stated within this sentence. The message of the manuscript would be more clearly summarized in the abstract if this sentence referred explicitly to the finding that anthropogenically derived SOM do not exhibit the complex thermodynamic behaviors of the biogenically derived SOM because of the absence of LLPS, instead of just describing the behavior as “different”.

More generally, the manuscript would benefit from more clear separation of what is a macroscopically observed CCN discrepancy that the authors are trying to resolve (humidity dependent hygroscopicity and size-dependent CCN activity) and what is an observed micro-scale phenomenon, the presence of which can explain these discrepancies (LLPS).

In terms of whether the manuscript could be shortened to aid communication of the most important finding, I consider the paragraph 304-312 to be insufficiently well-linked to the major findings of this work to justify its inclusion. If the author is suggesting that the model presented here could be applied to inorganic-organic mixed aerosol particles and SOM with inorganic seeds, this needs to be done more explicitly, and to include it as a suggestion for future work building on the work presented here.

Whilst the finding here are mostly presented without exaggeration, I take issue with the claim that the four substances here are representative of the majority of biogenically and anthropogenically derived SOM. This needs either more substantiation, or to more explicitly state the limitations of using only four compounds.

In general, previous literature on this subject is thoroughly covered and the manuscript places the current study very well within the context of ongoing developments in the field. A reference is missing for the first sentence of the main body of text (lines 39-40).

Some aspects of the methodology are not described in sufficient detail for the experiment to be reproduced.

Line 327 describes Toluene- and dodecane-derived SOMs being “primarily” produced from photooxidation by hydroxyl radicals. This implies they were also produced by other methods, which are not described. Data from Toluene- and dodecane-derived SOMs that are produced from different methods should be identified.

The dry diameter is used (I assume this is the mobility diameter from the DMA, this should be clearly stated), but detail is lacking about how the particles were dried and verification of the RH achieved by this drying method. A supplementary figure of the experimental set-up might aid communication of the methodology, or at least a fuller explanation in the text.

Reviewer #3 (Remarks to the Author):

Review of manuscript “Resolving the Mechanisms of Hygroscopic Growth and Cloud Condensation Nuclei Activity for Organic Particulate Matter” by Liu et al.

The hygroscopic growth and CCN activation characteristics are studied for organic aerosol particles. High precision measurements of the hygroscopic growth and CCN activity for secondary organic matter originating from dark ozonolysis of alpha-pinene and limonene show a) humidity-dependent changes of the hygroscopicity, b) liquid-liquid phase separation (LLPS) into a hydrophobic and a hydrophilic phase at high humidities ($>0,95\%$ RH for alpha-pinene derived particles), and c) the critical supersaturation for CCN activation is lowered compared to the expectation for a single-phase system. The three effects are consistently explained by a thermodynamic model describing the non-ideal mixing of water with the hydrophobic and hydrophilic components that constitute the two liquid phases. For LLPS, the hydrophobic phase is found to form the outer shell, lowering the surface tension, and therefore a lower critical supersaturation for CCN activation is observed than would be expected for single-phase particles.

The paper is very clearly written and concise. The key messages, the data and the resulting conclusions are presented in an illustrative and convincing way, and the presentation is well accessible also to non-specialists. I enjoyed reading the manuscript. From the combination of the precise laboratory studies with the LLPS model a detailed mechanistic understanding is achieved representing a substantial advancement of our fundamental understanding of hygroscopicity and CCN activity. There are just two issues that may preclude publication in a high-impact journal like Nature Communication:

1) Biogenic vs. anthropogenic precursors: The authors study four substances (alpha-pinene, limonene, toluene and dodecane). These are grouped as representative for biogenic and anthropogenic sources of secondary organic aerosol. Then it is demonstrated that the two biogenic substances show LLPS behavior while for the particles resulting from anthropogenic precursors LLPS is absent. The abstract, the main text and the figure captions continuously suggest that there is a difference between biogenic and anthropogenic organics in general that leads to LLPS for the biogenic precursors and not for the anthropogenic substances. E.g., in the abstract it is stated "For biogenically-derived organic PM ... humidity-dependent hygroscopicity, diameter-dependent CCN activity, and LLPS, were observed" and "For anthropogenically-derived organic PM, mechanistically different processes of CCN activation were observed because of the absence of LLPS". From testing just four substances out of the dozens or hundreds of biogenic and anthropogenic precursors for organic PM I am not convinced that this conclusion can be drawn. It is not demonstrated that the LLPS does apply only to particles from biogenic precursors, nor is it proven that it does not apply to other anthropogenic substances. While it is certainly worth noting that the studied monoterpenes are important biogenic VOCs while the other two substances are important precursors of PM from anthropogenic sources, it seems to me that the manuscript should be much more careful to avoid a generalization that cannot be made from studying such a limited number of substances.

2) Atmospheric relevance: Particles in the atmosphere that are large enough to act as CCN are rarely pure organic particles that result from just a single source such as alpha-pinene. Even small fractions of inorganic material such as ammonium sulfate or ammonium nitrate will influence the hygroscopic growth and the critical saturation values for activation strongly. Therefore it is not clear in how far the results can be transferred to atmospheric particles and in how far the effects of LLPS for certain organic precursors will actually influence hygroscopic behavior or CCN activation of particles in the atmosphere, especially when considering that the observed effects on κ and s_c are fairly subtle.

Editorial comments:

- SI, line before equation (S2): change " u_i " to " μ_i "

- First line of section S1.2: change "two liquid phase" to "two liquid phases"

- section S1.2, page 4: check wording of sentences "Partition of each compound..." and "The initial guess ... of a previous step..."

Reviewers' comments:**Reviewer #1 (Remarks to the Author):**

The manuscript entitled, "Resolving the Mechanisms of Hygroscopic Growth. . ." by Liu et al., focuses on resolving the long-standing question of why organic aerosols exhibit much greater apparent CCN activity under supersaturated conditions than would be predicted by measurements of growth factors conducted under subsaturated (RH<100%) environments. The authors present new measurements (and images) over a wide range of sub and supersaturated states for SOM of biogenic and anthropogenic origin. They also analyze the data using two models based upon the Flory Huggins Thermodynamic Model. The manuscript approaches an important problem for climate and atmospheric chemistry using new data and measurements. The manuscript is well written and the conclusions generally follow from the authors' analysis. I recommend the manuscript for publication when the authors have addressed the following in their revised manuscript.

We thank the reviewer for the careful reading of the manuscript and the associated questions and comments that were provided. These aspects are fully considered in the revision. Detailed replies to each comment are provided below.

1. The issue of LLPS has been discussed by numerous authors and only recently been applied in a robust way to CCN. For example, in Ref 21 (Ovadnevaite et al., 2017). The authors never really clarify how their modeling approach is different, the same or complimentary to this prior work. This work, which is only vaguely referred to, established the importance of LLPS at the point of activation (albeit) using a field data set. The authors should make it very clear how their modeling work connects with Ovadnevaite et al. In other words, is the model entirely new? or is this paper mainly about presenting new data to validate an already established modeling approach.

In the revised manuscript, we amended the introduction. Prior studies, including Ovadnevaite et al. (2017), are clearly discussed. We also added new descriptions for our model in the "Modeling Approach" section, and the connections between our method with prior work are further clarified. With these updates, the reader can recognize that our simplified thermodynamic scheme can utilize the new hygroscopicity data measured by the quartz crystal microbalance, and complicated information of organic molecular properties, as required by the model in Ovadnevaite et al. (2017), is not needed. The LLPS prediction, however, is similar to the approach employed in Ovadnevaite et al. (2017). Both predictions are rigorously calculated by minimizing the free energy.

The relevant revised text in the introduction reads as follows:

(Line 79-93) In the present study, LLPS, hygroscopic growth (for RH values from 10 to 95 %), and size-resolved CCN activation were investigated for laboratory-generated SOMs derived from different types of precursors representative of both biogenic and anthropogenic precursor sources. The data sets were interpreted in a solution-theory-based thermodynamic framework that considers the non-ideal mixing of water with hydrophobic and hydrophilic organic

compounds, including the possibility of facilitating LLPS and lowering surface tension. The thermodynamic parameters were constrained by the hygroscopic growth measurements obtained for a wide RH range using a quartz crystal microbalance, and the model predictions were compared with observed LLPS behaviors and size-resolved CCN activity. The solution-theory based thermodynamic framework used here is similar to the model framework used recently to successfully explained the observed high CCN number concentrations of organic/inorganic mixed particles in marine air masses and to established the importance of LLPS at the point of activation using a field data set [Ovadnevaite et al., 2017]. A major difference is that the current framework does not require detailed input of molecular properties, such as functional groups and molecular weights of organic compounds [Ovadnevaite et al., 2017; Zuend et al., 2010], which remains poorly constrained for atmospheric organic PM.

New descriptions in the Modeling Approach section are listed below.

(Line 172-190) **Modeling Approach.** *To explain the complex behaviors of water uptake and CCN activation observed for the biogenic SOMs, we developed a Flory-Huggins-Köhler model framework for SOM [Petters et al., 2006]. The model framework accounted for the interplay of non-ideal solution, LLPS, and surface activity. The treatment of LLPS and surface tension depression is conceptually similar to the model recently deployed by Ovadnevaite et al. [2017]. The thermodynamic framework herein, however, can use parameters directly obtained from the measurements, such as the interaction parameters and soluble fraction, and detailed input of organic molecular properties is not required.*

The Flory-Huggins solution theory is a thermodynamic model that takes an account of the dissimilarity in molecular sizes between small molecule solvents and large molecule solutes. The non-ideal mixing can be captured by including an interaction parameter, χ , which describes the enthalpy of mixing between organics and water (cf. Supporting Information). LLPS and phase equilibrium between two liquid phases can be calculated by minimizing Gibbs free energy following the method described in Zuend et al. [2010] Partitioning of water and organic components between shell and core phases is calculated for changing water activity values. The presence of LLPS is indicated by the phase equilibrium of two solutions of distinct chemical compositions. When LLPS occurs, surface tension is calculated using a semi-empirical approach considering complete or partial coverage of surface active organic monolayer, similar to the approaches used in Ovadnevaite et al.[2017] and Ruehl et al.[2016]

Two model configurations with different levels of complexity in the treatment of compositions were deployed in this study. The aim is to provide a multi-component framework with minimum complexity in organic composition that can still adequately describe the observed complex behaviors of hygroscopic growth, LLPS, and CCN activity.

2. Regarding surface tension (SI). The authors state in the SI that "In the present work, a σ value of 0.035 N m⁻¹ was used for both hydrophobic and hydrophilic organic components. The effective surface tension of a liquid phase was calculated as the volume-weighted mean of σ values of the components." It is not clear to my why 0.035 was selected? How sensitive are the results to this number, is this a fitting parameter? I would like to see some sensitivity analysis to better describe this assumption.

The σ value for pure organic component was chosen based-on previously reported literature values for organic acids. This is now clarified in the revised SI. Sensitivity analysis using different σ values is performed, and a new supplementary figure is added (Fig. S5). The results indicate that the selection of σ_{org} values within the typical range of 0.025 to 0.040 N m⁻¹ does not change the conclusions of the present work.

(SI, S1.3) *At room temperature, the typical σ values for organic acids range from 0.025 to 0.040 N m⁻¹ [Álvarez et al., 1997; Chumpitaz et al., 1999; Riipinen et al., 2007]. In the present work, a σ_{org} value of 0.035 N m⁻¹ was used for both hydrophobic and hydrophilic organic components for simplicity.*

(SI, S1.3) *Sensitivity analysis shows that changing the σ_{org} value from 0.025 to 0.040 N m⁻¹ can change the modeled κ_{CCN} value maximum by up to 15% in Model II, which is comparable to the uncertainty of the CCN measurements (Fig. S5). Changing the σ_{org} value does not alter the modeled κ_{CCN} in Model I, because surface concentration of organic solute gets strongly diluted at the condition of CCN activation.*

Also why was volume weighted σ values used instead of surface weighted? Volume weighting can substantially under account for surface active species (i.e. surfactants).

The assumption of using volume weighted σ values is further discussed in the revised manuscript.

(SI, S1.3) *The effective surface tension of a liquid phase was calculated as the volume-weighted mean of σ values of the components. In the case of no LLPS, this treatment is equivalent to an assumption of no bulk-surface partitioning, i.e., the surface concentration of organic solute is the same as the bulk concentration, and the effective surface tension is usually close to that of pure water at the point the CCN activation. In the presence of LLPS, however, surface-active, hydrophobic organic compounds are concentrated at the droplet surface, which substantially reduces the surface tension.*

3. Line 152-158: Here the authors rule out the gradual dissolution of sparingly soluble compounds as a way to reconcile sub and supersaturated kappa. They point to Fig. 1F as evidence. Given the importance of this conclusion, the authors should expand this discussion giving the reader a clear idea of the logic in addition to pointing to the figure.

The discussion is expanded in the revised manuscript and the new text reads as below.

(Line 148-155) *This interpretation, however, in part appears to be contradictory with the size-resolved CCN observation shown in Fig. 1D. For particles with an identical chemical composition, Köhler theory predicts that the dilution ratio at the activation point decreases with an increasing s_c (i.e., a decreasing $d_{m,dry}$). In the solubility-based formulation, a decreasing dilution ratio would result in less organic solute dissolved in the aqueous phase, therefore decreases effective κ value. This prediction is inconsistent with the size-resolved CCN measurements showing that the effective κ value increases with an increasing s_c (Fig. 1D).*

4. Model 1 (lines 184). It is not clear what the implications are for treating SOM as a surfactant? Does Model 1 allow SOM to lower water activity (or just surface tension). How is surface partitioning computed in this case. This should be clarified.

In the revised manuscript, the implications of the surface tension treatment are further clarified in several places.

(Main text, Line 205-208) *The pure-component SOM was treated as having a surface tension value lower than that of water. In the absence of LLPS, however, the surface concentration of organic solute was assumed to be equal to that of the bulk solution. The SOM mainly contributed to the CCN activity by the Raoult solute effect.*

(SI text, S1.3) *The effective surface tension of a liquid phase was calculated as the volume-weighted mean of σ values of the components. In the case of no LLPS, this treatment is equivalent to an assumption of no bulk-surface partitioning, i.e., the surface concentration of organic solute is the same as the bulk concentration, and the effective surface tension is usually close to that of pure water at the point the CCN activation. In the presence of LLPS, however, surface-active, hydrophobic organic compounds are concentrated at the droplet surface, which substantially lower the surface tension.*

5. Lines 214-220 The authors say the volume of the shell phase was insensitive to changes in water activity (unlike the core). Is this an observation or a constraint in the model? In other words, does the model allow for water uptake into the shell and re-partitioning of material from the shell to core or core to shell as water activity changes? The authors should clarify this. If this model doesn't allow for dynamic re-partitioning between shell and core, the authors should defend this model simplification.

The volume of the shell phase is calculated in the model by assuming phase equilibrium, meaning water uptake into the shell and dynamic re-partitioning between shell and core are always allowed for the changes in water activity. The phase equilibrium is rigorously solved in the model based on first principle (i.e., minimization of the free energy) with simplified chemical composition. This point is clarified in the revised manuscript.

(Line 228-229) *Once formed, the volume of the shell phase, as calculated in the model by assuming phase equilibrium, was insensitive to water activity, while the volume of the core grew by orders of magnitude at higher water activity (Fig. 3A).*

(Line 183-185) *LLPS and phase equilibrium between two liquid phases can be calculated by minimizing Gibbs free energy following the method described in Zuend et al. [2010] Partitioning of water and organic components between shell and core phases is calculated for changing water activity. The presence of LLPS is indicated by the phase equilibrium of two solutions of distinct chemical compositions.*

6. The Model (three components water, hydrophobic, and water soluble species) is clearly a

simplification for the complexity of SOM, which in principle is ok and necessary. Some discussion in the manuscript of this simplification is needed with appropriate caveats. For example, what could be the role that small populations of extremely surface active species play and could these alter the main conclusions of the paper?

This is a good point. We completely agree that our model is a simplification for the complexity of SOM. In the paper, we show that the binary model (water and organics) is a case of over-simplification, as it fails to predict the LLPS and the size-dependent CCN activity. The ternary model (water, hydrophobic, and hydrophilic species), however, appears to be the simplest system that can still capture the most important features of the measurements. This point is discussed in the revised manuscript.

(Line 191-195) Two model configurations with different levels of complexity in the treatment of compositions were deployed in this study. The aim is to provide a multi-component framework with minimum complexity in organic composition, which possibly can be obtained using experimental data and that can still adequately describe the observed complex behaviors of hygroscopic growth, LLPS, and CCN activity.

The ternary model, although simplified, indeed predicts that a small fraction (5-7% by mass) of hydrophobic, surface active organic species can reduce surface tension and play an important role in CCN activation. This point is further clarified in the manuscript.

(Line 309) The experimental and modeling results highlight the important role of a small fraction of hydrophobic organic compounds in water uptake and CCN activation.

7. Lines 64-67 The role of larger than expected droplet diameters has been observed in both supersaturated regimes (Ref 20) but also by Ruehl et al. in sub saturated conditions where the role of monolayer structure and associated changes in surface tension was explicitly investigated (Journal of Physical Chemistry A 118 (22), 3952-3966). It seems appropriate to include this reference in this part of the discussion.

The reference is added. The relevant sentence reads as follows:

(Line 40-44) A laboratory study suggested the formation of an organic surface film for submicrometer particles of organic/inorganic mixtures at high RH [Ruehl and Wilson, 2014]. The organic film and associated surface tension depression can explain the large droplet activation diameters at supersaturation observed for particles composed of SOM and ammonium sulfate [Ruehl et al., 2016].

Reviewer #2 (Remarks to the Author):**Review of Liu et al., Resolving the Mechanism of Hygroscopic Growth and Cloud Condensation Nuclei Activity for Organic Particulate Matter**

This paper claims that biogenically derived SOM exhibit non-ideal behavior in the form of LLPS, RH dependent hygroscopicity and discontinuous hygroscopicity at the point of water vapor saturation. It claims that this behavior is caused by non-ideal mixing of water with hydrophobic and hydrophilic organic components of the biogenically derived SOM, which are not present in anthropogenically derived SOM (which exhibit ideal behavior).

The non-ideal behavior of biogenically derived SOM has been observed by others, as pointed out on line 63 of the paper, as has LLPS in these specific SOMs (see figures 1 and 2). The novelty of this paper lies in obtaining size resolved CCN measurements and hygroscopicity data below 40% RH, and using these to parameterize and test a model to explain the observed behavior.

This paper will be of interest to others in the field as CCN activity is one of the larger uncertainties on aerosol-cloud-interactions and their effect on climate, and improved K values are needed to address this.

This paper challenges other ideas that have been proposed to explain enhanced CCN activity relative to predictions based on K values and provides a model/theoretical framework that could be tested on other SOM as a new way to understand CCN activity of SOM.

Convincing evidence that the proposed model, that considers polarity and mixing enthalpy with water and other organic molecules describes the observed behavior better than a model which describes an effective binary mixture of SOM and water, treating SOM as a surfactant.

We acknowledge the reviewer for the constructive comments that can improve the manuscript. These comments are incorporated in the revised manuscript. The reviewer's questions are addressed below.

8. While testing of this model on further biogenically and anthropogenically derived SOM would clearly be beneficial in determining its general utility, I do not consider that necessary for the publication of this work, provided this is properly commented on in the manuscript. Current lines 320-321 indicate the need of 'improvement' and 'better understanding of the physical and chemical properties of these compounds at a molecular level'. This falls short of a clear statement of the limitations of using only 4 compounds in this study and lacks clarity regarding what "improvements" the author considers necessary.

We have carefully revised manuscript and clearly mentioned the need of further studies for SOMs from other precursors.

(Line 341-343) *This study investigated four SOMs, and the properties of SOMs derived from other biogenic and anthropogenic precursor sources should be further investigated to examine the further generalization of the results.*

The unclear statement is revised. The updates are as follows:

(Line 345-346) *Future laboratory and field studies are needed for improvement of molecular characterization and better understanding of the physical and chemical properties of these compounds.*

9. The manuscript lack clarity, mostly in how the figures are described and presented and in the structure of the figures themselves. The first figure referred to is figure 1C, whereas it is customary to order figures by order of discussion in the main text. The order problem comes up again for figure 3. It is perhaps unclear to me panels a and b of figures 1 and 3 need to be grouped with panels C-F, and navigation of the paper might be improved by putting these as separate figures.

We thank the review for this helpful suggestion. The panels A and B of Figure 1 have been moved into a separate figure in the revised manuscript (new Fig. 2), as suggested by the reviewer. Similarly, we split Figure 3 into two separate figures (new Fig. 4 and Fig. 5). The order of new figures is consistent with the order of discussion in the revised manuscript.

10. The legend for figure 1 panels C-F is not well placed. I found myself searching for it when looking first at panel C. It is unclear whether the dashed orange and green lines representing model 1 are just hidden behind the solid lines of model 2 in panels C and D, or nor included, or indeed if these lines are the model or just a fit. Figure 2 panel a would benefit from an explanation of the drop off/increase in V/V_{dry} for the shell and core phases respectively as the water activity approaches unity.

We placed the legend of panels C-F (new Figure 1 in the revised manuscript) on the top of the entire figure.

Dashed lines of model 1 are hidden behind the solid lines of Model II, and this point is clarified in the figure caption.

(Figure 1, caption) *In panel A, dashed lines of Model I are hidden behind the solid lines of Model II because two models yield identical results in the absence of LLPS ($RH < 95\%$).*

The change of V/V_{dry} as the water activity approaches unity is also explained in the figure caption.

(Figure 3, caption) *The V/V_{dry} value of the shell phase drops off when water activity approaches unity because of the mixing of two phases in single phase.*

11. Similarly, the discontinuity in the hydrophobic SOM volume fraction in panel B at high water activity should be explained in the figure caption. The QCM measurements in panel C of figure 2 are not explained or referred to in the figure caption, and ought to be. The

cartoons of particle phases in figure 2 panels A and D need reference to and explaining in the caption.

The figure caption is revised as suggested.

(Figure 3, caption) *The discontinuity of hydrophobic SOM volume fraction shown in panel (B) represent onset of the LLPS.*

(Figure 3, caption) *The volume fractions of water calculated based on the mass-based QCM measurements of hygroscopic growth are shown in panel (C) for comparison.*

(Figure 3, caption) *Cartoons in panels A and D show the possible morphology of the particle. Blue and green colors represent water and organics, respectively.*

12. Clarity is also impeded by clumsy sentence structure of missing clauses, including the following:

In the abstract, line 32, the description of the behavior as “complex”, especially as this complexity has already been remarked upon in line 30 is redundant, and I suggest the word be omitted.

The change is made.

13. Line 55 would read better with “the negligible influence” rather than “a negligible influence”.

The change is made.

14. Line 261: “are valuable for elucidating’ could be just “elucidate”

The change is made.

15. Line 263: “improved the capability of water uptake measurements” – does this mean makes the measurements possible? Or more accurate? Or more precise? It could be better explained.

This is further clarified. The relevant sentence reads as below.

(Line 277-278) *The high sensitivity of the QCM method substantially improves the **precision and accuracy** of water uptake measurements in the low RH region.*

16. Line 320: “future laboratory and field studies are needed for improvement ... of the physical and chemical properties of these compounds”, needs to be better explained. Improvement in what sense?

This is further clarified. Please refer to the response to question 8.

17. The abstract does not currently provide a clear enough overview of the manuscript because of a failure to adequately explain some of the concepts introduced:

Line 28 states that the paper presents results of LLPS, hygroscopic growth and size resolved CCN activation studies, but a clear explanation of why LLPS is relevant for understanding “the mechanisms of hygroscopic growth and CCN activation” is both warranted and currently lacking.

We revised the abstract. Clarifications for the relevance of LLPS to hygroscopic growth and CCN activation are included in the updated abstract.

(Abstract) These behaviors can be explained by the non-ideal mixing of water with hydrophobic and hydrophilic organic components. The non-ideality-driven LLPS further enhanced water uptake and induces lowered surface tension at high relative humidity, which resulted in a lower barrier to CCN activation.

18. Line 35 – it needs to be more clearly stated that it is alternative mechanism to explain humidity dependent hygroscopicity and size-dependent CCN activity that are being ruled out.

We agree that further clarification of the “alternative mechanism” would be helpful. However, due to the word limitation of Nature Communications (150 words for the abstract), we have removed this unclear sentence in the revised abstract. The relevant content has been discussed in the main text.

19. Lines 35-36, “For anthropogenically-derived organic PM, mechanistically different processes of CCN activation were observed because of the absence of LLPS” is ambiguous. What these processes are different to should at least be clearly stated within this sentence. The message of the manuscript would be more clearly summarized in the abstract if this sentence referred explicitly to the finding that anthropogenically derived SOM do not exhibit the complex thermodynamic behaviors of the biogenically derived SOM because of the absence of LLPS, instead of just describing the behavior as “different”.

To clarify, the relevant sentence is re-written as suggested.

(Abstract) Secondary organic material representing anthropogenic sources did not exhibit complex thermodynamic behaviors because of the absence of LLPS.

20. More generally, the manuscript would benefit from more clear separation of what is a macroscopically observed CCN discrepancy that the authors are trying to resolve (humidity dependent hygroscopicity and size-dependent CCN activity) and what is an observed micro-scale phenomenon, the presence of which can explain these discrepancies (LLPS).

The reviewer's point is well taken and the revised text specifically identifies macroscopically observed CCN/hygroscopicity discrepancy and microscopically observed LLPS.

(Line 306-308) *The simplified ternary mixture model considering water, hydrophobic species, and hydrophilic species can bridge the macroscopically observed discrepancy between CCN activity and hygroscopicity with microscopically observed LLPS.*

In addition, the figures are revised to call out this contrast. The results of macroscopic observations (CCN activity and hygroscopic growth) and microscopic observation (LLPS) are organized in separate figures. Please also refer to response to question #9.

21. In terms of whether the manuscript could be shortened to aid communication of the most important finding, I consider the paragraph 304-312 to be insufficiently well-linked to the major findings of this work to justify its inclusion. If the author is suggesting that the model presented here could be applied to inorganic-organic mixed aerosol particles and SOM with inorganic seeds, this needs to be done more explicitly, and to include it as a suggestion for future work building on the work presented here.

This paragraph is completely rewritten, and new clarifications are added.

(Line 322-327) *As a caveat, the model framework employed herein only considered interactions between organic solutes and water. This treatment is appropriate for the present work because the generated SOM was free of inorganic inclusions. Particles in the atmosphere, however, are usually internal mixtures of organic and inorganic species. In this case the inorganic species can cause salting out, resulting in increased LLPS [Marcolli and Krieger, 2006]. Ionic interactions of electrolyte with water and non-electrolytes should be included in the model in future studies.*

22. Whilst the finding here are mostly presented without exaggeration, I take issue with the claim that the four substances here are representative of the majority of biogenically and anthropogenically derived SOM. This needs either more substantiation, or to more explicitly state the limitations of using only four compounds.

We have carefully revised the manuscript and clearly state that the results are based on studied four SOMs, and that future research is needed to test the further generalization of the results.

(Line 341-343) *This study investigated four SOMs, and the properties of SOMs derived from other biogenic and anthropogenic precursor sources should be investigated to examine the further generalization of the results.*

23. In general, previous literature on this subject is thoroughly covered and the manuscript places the current study very well within the context of ongoing developments in the field. A reference is missing for the first sentence of the main body of text (lines 39-40).

The reference is added for the first sentence.

24. Some aspects of the methodology are not described in sufficient detail for the experiment to be reproduced.

Line 327 describes Toluene- and dodecane-derived SOMs being “primarily” produced from photooxidation by hydroxyl radicals. This implies they were also produced by other methods, which are not described. Data from Toluene- and dodecane-derived SOMs that are produced from different methods should be identified.

The original sentence was unclear, and it is re-written in the revised manuscript.

(Line 353-354) *Toluene- and dodecane-derived SOMs were produced in the OFR by photooxidation of the precursors primarily by hydroxyl radicals.*

25. The dry diameter is used (I assume this is the mobility diameter from the DMA, this should be clearly stated), but detail is lacking about how the particles were dried and verification of the RH achieved by this drying method. A supplementary figure of the experimental set-up might aid communication of the methodology, or at least a fuller explanation in the text.

In the revised manuscript, new descriptions of the experimental methodology, including definition of the dry diameter, drying method, and RH verification for the CCN measurement are added. We also add other details of experimental methodology and data analysis that can help the reader to reproduce our work. We also make clear reference to previous publications that schematic diagrams of experiment can be found.

(Line 366-392) **Quartz-Crystal Microbalance.** *Thin films of SOM were grown by electrostatic precipitation of aerosol particles onto SiO₂-coated QCM crystals (Q-sense QSX 303) using a Nanometer Aerosol Sampler (TSI 3089)[Liu et al., 2016; Liu et al., 2013]. Film thicknesses was on the order of 100 nm. The SOM-laden QCM crystals were mounted into a temperature- and humidity-controlled flow module. Different relative humidity values, as continuously monitored by an RH sensor (Rotronic, HydroClip 2), were achieved by changing the mixing ratio of dry and humidified nitrogen flows using two mass flow controllers (MKS M100b), and the total flow rate was kept at 20 cm³ min⁻¹. Film mass was continuously monitored by the QCM (Q-sense E4). A schematic diagram of the QCM measurement can be found in a previous publication [Liu et al., 2016].*

Cloud Condensation Nuclei Counter. *Polydisperse SOM particles produced from the OFR were size-selected by a differential mobility analyzer (DMA; TSI 3081). The particles were dried using a Nafion dryer (Perma Pure Inc.), and the RH prior to entering the CCN counter was kept below 20%, as confirmed by an RH probe (Omega, RH-USB). The CCN activation ratio was determined as a function of particle dry mobility diameter from simultaneous measurements of a continuous flow CCN counter (DMT CCNC) and a condensation particle counter (CPC; TSI 3772). Supersaturation in the CCNC varied between 0.15% and 1%. Calibration was based on the activation of ammonium sulfate particles. Critical dry diameter $d_{m,dry}$ for each*

supersaturation value was determined from fitting of the activation curve to a Gauss error function. The doubly charged particle fraction was subtracted prior to data fitting. The detailed protocol of size-resolved CCN calibration and measurement followed that described in Rose et al [Rose et al., 2008].

Reviewer #3 (Remarks to the Author):**Review of manuscript “Resolving the Mechanisms of Hygroscopic Growth and Cloud Condensation Nuclei Activity for Organic Particulate Matter” by Liu et al.**

The hygroscopic growth and CCN activation characteristics are studied for organic aerosol particles. High precision measurements of the hygroscopic growth and CCN activity for secondary organic matter originating from dark ozonolysis of alpha-pinene and limonene show a) humidity-dependent changes of the hygroscopicity, b) liquid-liquid phase separation (LLPS) into a hydrophobic and a hydrophilic phase at high humidities (>0,95% RH for alpha-pinene derived particles), and c) the critical supersaturation for CCN activation is lowered compared to the expectation for a single-phase system. The three effects are consistently explained by a thermodynamic model describing the non-ideal mixing of water with the hydrophobic and hydrophilic components that constitute the two liquid phases. For LLPS, the hydrophobic phase is found to form the outer shell, lowering the surface tension, and therefore a lower critical supersaturation for CCN activation is observed than would be expected for single-phase particles.

The paper is very clearly written and concise. The key messages, the data and the resulting conclusions are presented in an illustrative and convincing way, and the presentation is well accessible also to non-specialists. I enjoyed reading the manuscript. From the combination of the precise laboratory studies with the LLPS model a detailed mechanistic understanding is achieved representing a substantial advancement of our fundamental understanding of hygroscopicity and CCN activity. There are just two issues that that may preclude publication in a high-impact journal like Nature Communication:

We thank the reviewer for careful reading of the manuscript and for acknowledging the merit of the present manuscript. The questions and comments are fully considered in the revised manuscript. Detailed replies are listed below.

26. Biogenic vs. anthropogenic precursors: The authors study four substances (alpha-pinene, limonene, toluene and dodecane). These are grouped as representative for biogenic and anthropogenic sources of secondary organic aerosol. Then it is demonstrated that the two biogenic substances show LLPS behavior while for the particles resulting from anthropogenic precursors LLPS is absent. The abstract, the main text and the figure captions continuously suggest that there is a difference between biogenic and anthropogenic organics in general that leads to LLPS for the biogenic precursors and not for the anthropogenic substances. E.g., in the abstract it is stated “For biogenically-derived organic PM ... humidity-dependent hygroscopicity, diameter-dependent CCN activity, and LLPS, were observed” and “For anthropogenically-derived organic PM, mechanistically different processes of CCN activation were observed because of the absence of LLPS”. From testing just four substances out of the dozens or hundreds of biogenic and anthropogenic precursors for organic PM I am not convinced that this conclusion can be drawn. It is not demonstrated that the LLPS does apply only to particles from biogenic precursors, nor is it proven that it does not apply to other anthropogenic substances. While it is certainly worth noting that the studied monoterpenes are important biogenic VOCs

while the other two substances are important precursors of PM from anthropogenic sources, it seems to me that the manuscript should be much more careful to avoid a generalization that cannot be made from studying such a limited number of substances.

To response to this comment and the relevant comment 22, we made changes through the manuscript to avoid over-generalization of the results. For example, the relevant sentence in the revised abstract reads as follows:

(Abstract) Herein, complex thermodynamic behaviors, including humidity-dependent hygroscopicity, diameter-dependent CCN activity, and liquid-liquid phase separation (LLPS) were observed in laboratory for biogenically derived secondary organic material representative of similar atmospheric organic PM.

Additional revisions are made in the abstract:

*(Line 412-416) For SOMs representative of **monoterpene-derived** biogenic organic PM, the non-ideality and associated LLPS were observed, which can have an important role in determining hygroscopicity and CCN activity. By comparison, no LLPS was observed for SOMs representative of **anthropogenic organic PM derived from aromatic and alkane precursors**, and the associated hygroscopicity and CCN activity were well described by the κ -model.*

We also clearly state that the results are based on four SOMs and further studies are needed to test the further generalization. Please refer to the response to question #22 for revised text.

27. Atmospheric relevance: Particles in the atmosphere that are large enough to act as CCN are rarely pure organic particles that result from just a single source such as alpha-pinene. Even small fractions of inorganic material such as ammonium sulfate or ammonium nitrate will influence the hygroscopic growth and the critical saturation values for activation strongly. Therefore it is not clear in how far the results can be transferred to atmospheric particles and in how far the effects of LLPS for certain organic precursors will actually influence hygroscopic behavior or CCN activation of particles in the atmosphere, especially when considering that the observed effects on kappa and s_c are fairly subtle.

This is a good point. New discussions on the effect of inorganic inclusions on LLPS, and future work to expand current model framework are added.

(Line 322-327) As a caveat, the model framework employed herein only considered interactions between organic solutes and water. This treatment is appropriate for the present work because the generated SOM was free of inorganic inclusions. Particles in the atmosphere, however, are usually internal mixtures of organic and inorganic species. In this case the inorganic species can cause salting out, resulting in increased LLPS. Ionic interactions of electrolyte with water and non-electrolytes should be included in the model in future studies.

Editorial comments:

- SI, line before equation (S2): change " u_i " to " μ_i "
- First line of section S1.2: change "two liquid phase" to "two liquid phases"
- section S1.2, page 4: check wording of sentences "Partition of each compound..." and "The initial guess ... of a previous step..."

The corrections are made.

References

- Álvarez, E., G. Vázquez, M. Sánchez-Vilas, B. Sanjurjo, and J. M. Navaza (1997), Surface Tension of Organic Acids + Water Binary Mixtures from 20 °C to 50 °C, *Journal of Chemical & Engineering Data*, 42(5), 957-960, doi:10.1021/je970025m.
- Chumpitaz, L. D. A., L. F. Coutinho, and A. J. A. Meirelles (1999), Surface tension of fatty acids and triglycerides, *Journal of the American Oil Chemists' Society*, 76(3), 379-382, doi:10.1007/s11746-999-0245-6.
- Liu, P., Y. J. Li, Y. Wang, M. K. Gilles, R. A. Zaveri, A. K. Bertram, and S. T. Martin (2016), Lability of secondary organic particulate matter, *Proc Natl Acad Sci USA*, 113(45), 12643-12648, doi:10.1073/pnas.1603138113.
- Liu, P., Y. Zhang, and S. T. Martin (2013), Complex refractive indices of thin films of secondary organic materials by spectroscopic ellipsometry from 220 to 1200 nm, *Environ Sci Technol*, 47(23), 13594-13601, doi:10.1021/es403411e.
- Marcollì, C., and U. K. Krieger (2006), Phase Changes during Hygroscopic Cycles of Mixed Organic/Inorganic Model Systems of Tropospheric Aerosols, *J Chem Phys A*, 110(5), 1881-1893, doi:10.1021/jp0556759.
- Ovadnevaite, J., et al. (2017), Surface tension prevails over solute effect in organic-influenced cloud droplet activation, *Nature*, 546, 637, doi:10.1038/nature22806
<https://www.nature.com/articles/nature22806#supplementary-information>.
- Petters, M. D., S. M. Kreidenweis, J. R. Snider, K. A. Koehler, Q. Wang, A. J. Prenni, and P. J. Demott (2006), Cloud droplet activation of polymerized organic aerosol, *Tellus B*, 58(3), 196-205, doi:10.1111/j.1600-0889.2006.00181.x.
- Riipinen, I., I. K. Koponen, G. P. Frank, A.-P. Hyvärinen, J. Vanhanen, H. Lihavainen, K. E. J. Lehtinen, M. Bilde, and M. Kulmala (2007), Adipic and Malonic Acid Aqueous Solutions: Surface Tensions and Saturation Vapor Pressures, *J Chem Phys A*, 111(50), 12995-13002, doi:10.1021/jp073731v.
- Rose, D., S. S. Gunthe, E. Mikhailov, G. P. Frank, U. Dusek, M. O. Andreae, and U. Pöschl (2008), Calibration and measurement uncertainties of a continuous-flow cloud condensation nuclei counter (DMT-CCNC): CCN activation of ammonium sulfate and sodium chloride aerosol particles in theory and experiment, *Atmos Chem Phys*, 8(5), 1153-1179, doi:10.5194/acp-8-1153-2008.
- Ruehl, C. R., J. F. Davies, and K. R. Wilson (2016), An interfacial mechanism for cloud droplet formation on organic aerosols, *Science*, 351(6280), 1447-1450, doi:10.1126/science.aad4889.

- Ruehl, C. R., and K. R. Wilson (2014), Surface Organic Monolayers Control the Hygroscopic Growth of Submicrometer Particles at High Relative Humidity, *J Chem Phys A*, 118(22), 3952-3966, doi:10.1021/jp502844g.
- Zuend, A., C. Marcolli, T. Peter, and J. H. Seinfeld (2010), Computation of liquid-liquid equilibria and phase stabilities: implications for RH-dependent gas/particle partitioning of organic-inorganic aerosols, *Atmos Chem Phys*, 10(16), 7795-7820, doi:10.5194/acp-10-7795-2010.

Reviewer #1 (Remarks to the Author):

The authors have adequately responded to the issues I raised in the first review. I also judge that they have made a substantial effort to address the other reviewers comments and criticisms.

Therefore I recommend publication of the manuscript in its current form.

Reviewer #2 (Remarks to the Author):

All points raised in the previous review were satisfactorily addressed with one exception. The point labelled 24 took issue with the description of SOM being 'primarily' produced by photooxidation by hydroxyl radicals. The use of 'primarily' suggests there were also other oxidants, but these are not discussed. The revised sentence does a much better job of describing the method, but still includes this use of primarily. If other oxidants are unknown, this should be stated outright, otherwise a full description should be given.

Reviewer #3 (Remarks to the Author):

I agree with the changes of the manuscript in response to the reviews. I have no further comments.

Technical correction: l. 112 and 113 please correct „explained“ and „established“

Reviewers' comments:**Reviewer #2 (Remarks to the Author):**

All points raised in the previous review were satisfactorily addressed with one exception. The point labelled 24 took issue with the description of SOM being ‘primarily’ produced by photooxidation by hydroxyl radicals. The use of ‘primarily’ suggests there were also other oxidants, but these are not discussed. The revised sentence does a much better job of describing the method, but still includes this use of primarily. If other oxidants are unknown, this should be stated outright, otherwise a full description should be given.

Further description and an additional reference are added in the revised manuscript:
(Methods) *Toluene- and dodecane-derived SOMs were produced in the OFR by photooxidation of the precursors primarily by hydroxyl radicals. Non-OH pathways, such as the photolysis of VOC precursors, were estimated to account for less than 1% of precursor loss [Peng et al., 2016].*

Reviewer #3 (Remarks to the Author):

I agree with the changes of the manuscript in response to the reviews. I have no further comments.

Technical correction: l. 112 and 113 please correct “explained” and “established”

The corrections are made.

References:

Peng, Z., D. A. Day, A. M. Ortega, B. B. Palm, W. Hu, H. Stark, R. Li, K. Tsigaridis, W. H. Brune, and J. L. Jimenez (2016), Non-OH chemistry in oxidation flow reactors for the study of atmospheric chemistry systematically examined by modeling, *Atmos Chem Phys*, 16(7), 4283-4305, doi:10.5194/acp-16-4283-2016.